# An Overview of Blockchain and IoT Integration for Secure and Reliable Health Records Monitoring

**Shadab Alam** [1] , **Surbhi Bhatia** [2,*] , **Mohammed Shuaib** [1,*] , **Mousa Mohammed Khubrani** [1] , **Fayez Alfayez** [3] , **Areej A. Malibari** [4] and **Sadaf Ahmad** [5]

[1] College of Computer Science & IT, Jazan University, Jazan 45142, Saudi Arabia
[2] Department of Information Systems, College of Computer Science and Information Technology, King Faisal University, P.O. Box 420, Al-Ahsa 31982, Saudi Arabia
[3] Department of Computer Science and Information, College of Science, Majmaah University, Al-Majmaah 11952, Saudi Arabia
[4] Department of Industrial and Systems Engineering, College of Engineering, Princess Nourah bint Abdulrahman University, P.O. Box 84428, Riyadh 11671, Saudi Arabia
[5] Department of Computer Science, Aligarh Muslim University, Aligarh 202001, India
* Correspondence: sbhatia@kfu.edu.sa (S.B.); msashraf@jazanu.edu.sa (M.S.)

**Abstract:** The Internet of Things (IoT) and blockchain (BC) are reliable technologies widely employed in various contexts. IoT devices have a lot of potential for data sensing and recording without human intervention, but they also have processing and security issues. Due to their limited computing power, IoT devices cannot use specialized cryptographic security mechanisms. There are various challenges when using traditional cryptographic techniques to transport and store medical records securely. The general public's health depends on having an electronic health record (EHR) system that is current. In the era of e-health and m-health, problems with integrating data from various EHRs, preserving data interoperability, and ensuring that all data access is in the patient's hands are all obstacles to creating a dependable EHR system. If health records get into the wrong hands, they could endanger the lives of patients and their right to privacy. BC technology has become a potent tool for ensuring recorded data's immutability, validity, and confidentiality while enabling decentralized storage. This study focuses on EHR and other types of e-healthcare, evaluating the advantages of complementary technologies and the underlying functional principles. The major BC consensus mechanisms for BC-based EHR systems are analyzed in this study. It also examines several IoT-EHR frameworks' current infrastructures. A breakdown of BC integration's benefits with the IoT-EHR framework is also offered. A BC-based IoT-EHR architecture has been developed to enable the automated sensing of patient records and to store and retrieve these records in a secure and reliable environment. Finally, we conduct a security study to demonstrate the security of our suggested EHR framework.

**Keywords:** healthcare; patient monitoring; EHR; blockchain; IoT; reliability; security

## 1. Introduction

IoT is the concept that everything should be connected to the internet. The software, sensors, actuators, and connectors that enable connections, data gathering, and data transmission between vehicles, home appliances, and other goods with embedded electronics are covered [1]. However, BC aims to preserve the infrastructure's dependability, immutability, and trustworthiness. A distributed database called BC has an encrypted ledger. A chain of blocks called a BC comprises several recently validated transactions. Cryptographical connections are made between every block. These transaction data are saved in each block, and the block is also given a consolidated hash code. A new block is appended to the BC whenever a transaction is completed, and the chain keeps expanding [2].

Two other sectors that have impacted public life are healthcare and EHR. The healthcare industry faces significant difficulty in managing and recovering the vast amount of personal health data generated by routine business and service operations. Wearables and other healthcare monitoring devices produce a ton of data about an individual's health. Most health data are unavailable, non-standardized across systems, and challenging to comprehend, use, and exchange. Due to the introduction of new technologies like IoT and BC, the healthcare industry has experienced exponential growth in recent years [3]. Adopting such technologies has enriched the healthcare segment in numerous spheres. IoT device proliferation increases the amount of information the internet processes, creating new security and privacy concerns. To unite these three technological sectors, little research or efforts have been made [4].

Security risks are more prominent in the medical industry and require specific care, even more so when IoT is involved. Article [5] reviews the various IoT-based healthcare systems and IoT application areas across multiple healthcare aspects, especially EHR. It highlights the issue of heterogeneity in IoT sensors while integrating them and further suggests applying the cloud architecture to resolve the heterogeneity and interoperability issues. Finally, it highlights the privacy issues that stem from the vulnerability of IoT and cloud systems. It suggests that traditional cryptographic techniques cannot be applied to IoT sensors due to resource constraints. A systematic review of significant research works in IoT applications in the healthcare domain has been carried out in [6]. The study highlights the various applications of IoT in healthcare and shows the security and privacy concerns.

Furthermore, it reviews the different cryptographic mechanisms to provide security to the IoT systems. It concludes that it is challenging to implement any suitable cryptographic mechanism to maintain security due to the heterogeneity of the devices. It also highlights the issue of centralized structure in the case of cloud-based IoT systems. A review of various healthcare IoT (HIoT) applications, advantages, and recent trends in the domain have been presented. It highlighted various challenges, including privacy and security concerns, and suggested possible solutions [7]. Paper [8] analyzes the security challenges in IoT-based traditional systems and further reviews the various security standards for healthcare data, like the Health Insurance Portability and Accountability Act (HIPAA) and its implications. Finally, it justifies the role of BC in resolving the security issue and standardization of EHR systems. A review of IoT-based healthcare applications has been conducted in [9], including various architectures such as cloud- and fog-based IoT healthcare systems. This study highlights major challenges like latency, fault tolerance, energy efficiency, security, and interoperability and the role of fog- and cloud-based IoT healthcare architectures.

These studies further highlight the weakness of IoT nodes in resource constraints; therefore, traditional cryptographic techniques are unsuitable. To resolve these issues, a cloud computing-based structure has been proposed to take care of processing tasks. Cloud-based systems are also susceptible to various types of security attacks due to their centralized nature [10,11]. Furthermore, implementing the requirements of regulations like HIPAA and GDPR cannot be achieved entirely using traditional systems, and non-compliance will attract heavy fines [12]. Identity management issues and providing users/patients control over their data are difficult in conventional centralized and cloud-based architecture [13]. Parallel to the advancements in the e-health arena, a new technology called BC is a peer-to-peer system that establishes worldwide consensus. It ensures that previously approved transactions cannot be altered or changed. While BC is a secure solution, it does have significant limitations, particularly when utilized with resource-constrained IoT devices. Medical data are extremely precious and must be handled with care in order to avoid data manipulation. In this view, BC offers numerous significant properties, including tamper proofing, immutability, traceability, data correctness, security, and anonymity, all without breaching the privacy of a third party [14]. Due to the intrinsic capabilities of BC, it is very much suitable for healthcare applications.

This work explores how BC operates on multiple platforms while suggesting that BC applications are inefficient for resource-constrained IoT devices. The new BC's fun-

damental weakness is that it uses computationally expensive operations unsuitable for resource-constrained systems. It can compromise some data privacy in return for decreased computing and energy usage, such as IoT [15]. A healthcare communication network is a mechanism that enables healthcare agencies such as doctors, nurses, patients, medications, laboratories, suppliers, and healthcare authorities to communicate with one another. Healthcare providers can introduce compliance mechanisms to protect organizational interactions [16]. BC encryption can be incorporated into their front-end networks to connect with doctors and nurses or their back-end systems for hosting electronic health records (EHRs). Patients can provide accurate, immutable reports and access to EHRs without communicating with care providers or treatment portals. BC can modify the way medical processes are performed. The volume of data produced by IoT devices is increasingly growing. E-health, or intelligent patient treatment, is one of the fascinating applications of IoT technologies. Any inappropriate access to medical data created by IoT devices is harmful [17]. Limited focus and a handful of studies have been carried out to incorporate all three areas to combine into one. This paper analyzes how well BC operates in several readily accessible platforms and suggests that complete BC operations are inefficient for IoT devices having limited resources [18]. The key issue in BC adaptation is its highly computing-intensive hashing operations that are not suitable for low-end devices and sensor devices with limited capabilities that share knowledge confidentiality levels for computing and energy savings. The consensus mechanism is BC applications' backbone and most resource-intensive phase [19].

This paper investigates multiple consensus mechanisms commonly used in all BC applications and discovers appropriate IoT networks to support electronic health record (EHR) systems and other healthcare services. A patient- and organization-driven BC and IoT health data processing system is presented. In the end, a new BC-based IoT-EHR framework has been offered to provide secure and reliable electronic health records with interoperability features. The key contributions of this paper are summarized as follows:

1. To summarize the IoT and BC applications in EHR;
2. To review the research and contributions applying IoT and BC in EHR;
3. To deliberate on and review existing BC consensus algorithms for the BC-based IoT-EHR applications;
4. To propose a BC-based IoT-EHR framework for secure and reliable health record storage supporting secure and reliable health record storage with interoperability features.

Several researchers have demonstrated BC's healthcare efficiency. Recent papers [18,20–23] analyzed existing work on healthcare BC technology to provide security. These related works have been summarized in Table 1. This paper reviews current works on integrating BC with IoT-EHR. Neither of these works have reviewed the consensus mechanism that is the core of any blockchain system. Its efficiency decides the outcome. For IoT-EHR systems, the standard consensus mechanisms cannot be considered, as they require high resource consumption that is generally unavailable in these environments. This paper reviews the prominent consensus mechanisms to analyze them on the parameters of IoT compliant, basic Concept, popularity, e-health support, adaptability, accessibility, and energy consumption to find the most suitable consensus mechanism for BC-based IoT EHR systems. We further propose a new BC-based IoT-EHR framework for secure and reliable EHR data transaction and storage that supports the interoperability of health records.

**Table 1.** Review of related works.

| Ref | Contribution | Year | BC | HC | IoT |
|---|---|---|---|---|---|
| [5] | Reviews the various IoT-based healthcare systems and IoT application areas across multiple healthcare aspects, especially EHR. | 2020 | N | Y | Y |
| [6] | The research highlights the various applications of IoT in healthcare and shows the security and privacy concerns. Further reviews the different cryptographic mechanisms to provide security to the IoT systems. | 2019 | N | Y | Y |
| [7] | A review of various HIoT applications, advantages, and recent trends in the domain. It also highlighted various challenges that include privacy and security concerns and suggested the possible solutions. | 2021 | N | Y | Y |
| [8] | Reviews the various security standards for healthcare data like Health Insurance Portability and Accountability Act (HIPAA) and its implications. | 2021 | N | Y | Y |
| [9] | Reviews various architectures that include cloud- and fog-based IoT healthcare systems. This study highlights significant challenges like latency, fault tolerance, energy efficiency, security, and interoperability and the role of fog- and cloud-based IoT healthcare architectures. | 2020 | N | Y | Y |
| [18] | Reviews the BC applications and BC technologies for healthcare. | 2019 | Y | Y | N |
| [20] | Review of applying BC technology in medical healthcare for protecting patient healthcare data, | 2019 | Y | Y | N |
| [21] | A comprehensive study of defining and assessing BC's use in healthcare and an analysis of its problems and advantages. | 2019 | Y | Y | N |
| [22] | A systematic survey of applying BC in healthcare applications that further analyzes and evaluates the adoption. | 2019 | Y | Y | N |
| [23] | Reviews many use cases for applying BC in healthcare. | 2019 | Y | Y | N |
| [24] | The authors explored BC healthcare applications. However, neither the issues nor the solutions were highlighted. | 2019 | Y | P | Y |
| [25] | The article concentrated on BC applications for the IoT. | 2019 | Y | N | Y |
| [26] | They discussed BC in cybersecurity but not specifically its applications in healthcare. | 2019 | Y | N | Y |
| [27] | Discussed the usage of BC-based patient identification in healthcare. | 2020 | Y | Y | N |
| [28] | The authors examined IoT-based healthcare systems, including possible uses, difficulties, and limitations. However, numerous recent studies have presented viable methods for using BC in healthcare, which is lacking from this research. | 2020 | Y | P | Y |

**Table 1.** *Cont.*

| Ref | Contribution | Year | BC | HC | IoT |
|---|---|---|---|---|---|
| [29] | A thorough examination of BC-based healthcare-related work conducted between 2016 and January 2020 was included in the research. As a result, a new review article highlighting current challenges and solutions is required. | 2020 | Y | Y | Y |
| [17] | The study focuses mainly on the potential and features of BC in healthcare data management. It did not stress how BC works or how it addresses the shortcomings of the current healthcare IT mechanisms. | 2021 | Y | Y | N |
| [30] | The study focuses on BC's applicability and problems in IoT. However, the writers did not address all of the major healthcare challenges. | 2021 | Y | P | Y |
| [13] | This study proposes a BC-based framework for Authentication, Authorization, and Audit in healthcare applications. It also reviews the issues with the traditional systems not implementing BC and the advantages of BC adoption in healthcare. It does not discuss the EHR aspects. | 2022 | Y | Y | Y |
| **Our paper** | This paper discusses the use of BC and the IoT in EHR systems and proposes a new framework | 2022 | Y | Y | Y |

BC—Blockchain, HC—Healthcare, Y—Yes, N—No, P—Partial.

## 2. Background Study

### 2.1. Electronic Health Record (EHR) System

EHR is a compilation of patients' electronic health information. Information associated with personal healthcare is stored in the personal health record (PHR). This information is retrieved from devices that can be worn and are controlled by patients. Patients can hand over their PHR information to healthcare professionals. Theoretically, the EHR mechanism aims to maximize the security of the stored data, upholding privacy and availability [31,32]. Furthermore, it ensures that data are only shared between authentic users, for instance, only allowing access to those medical professionals authorized to obtain any patient's electronic data to run their diagnosis. Figure 1 provides a general structure of the IoT-EHR system.

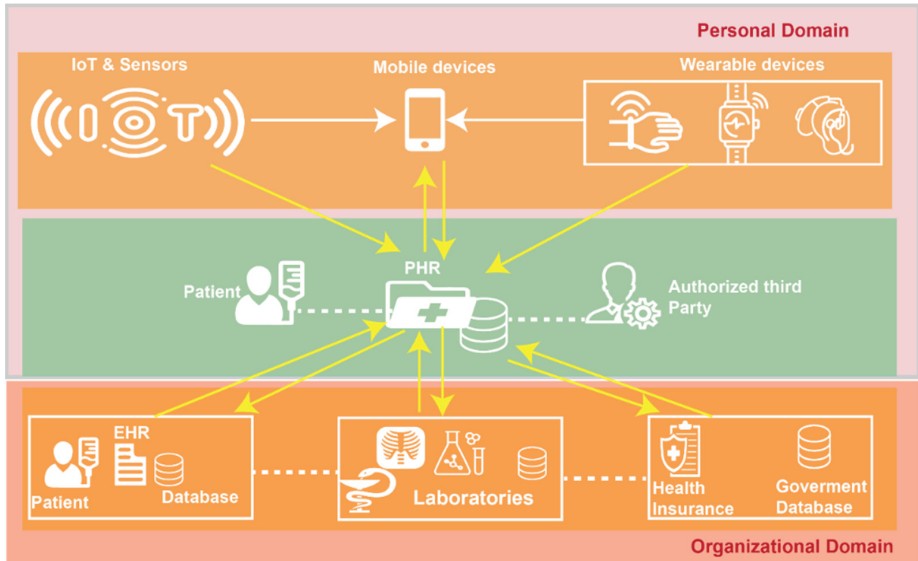

**Figure 1.** Structure of IoT-EHR system.

EHR is highly beneficial for machine learning and data analysis as it retains colossal amounts of data. It makes it instrumental for future research efforts focused on forecasting diseases, such as cases of COVID-19. Moreover, IoT and such wearable devices are pivotal in collecting relevant information and uploading it to EHR and PHR systems. It further adds to the facilitation of personalized healthcare services and healthcare monitoring [33].

### 2.2. Internet of Things (IoT)

Healthcare has faced several issues in recent decades as a result of rising healthcare costs, population expansion, and a shortage of caregivers. This scenario became more severe and crucial in recent years when the globe experienced a significant spread of COVID-19, resulting in, among other things, several challenges linked to exchanges and medical data management. A healthcare system primarily comprises hospital ward collaboration, medical diagnostic development, coordination across medical organizations, and collecting information about and from patients directly or through a network of linked devices and sensors.

IoT is essentially a system for linking devices, that is, the network of physical devices, items, or humans equipped with unique system identifiers (UIDs) and capable of transmitting data. Another aspect of the internet is that the things in the IoT are linked similarly to humans and computers, to which internet protocol addresses may be allocated and to which data can be sent across the network or to another man-made object. IoT technology is widespread and applied in each domain that requires data collection and sensing from different sources [8]. The major IoT applications have been summarized in Figure 2.

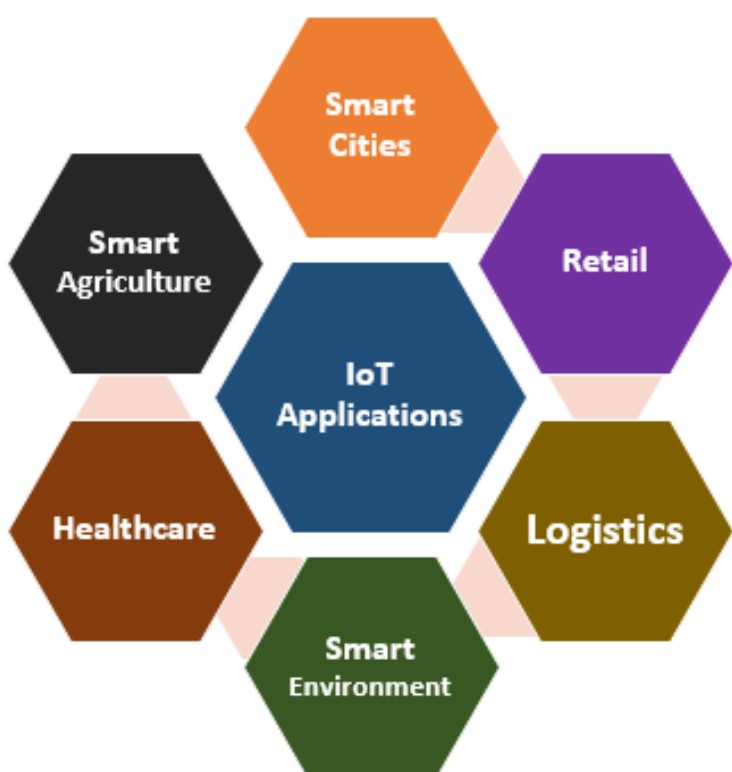

**Figure 2.** Applications of IoT.

### 2.3. Security Challenges Related to IoT in EHR

One major challenge for IoT devices is their security, particularly the end-to-end data security in any IoT domain. The concept of IoT devices enabling networking across various appliances and devices is relatively new; hence, security is not in-built into the design of IoT products. The issue also arises from assigning default or hardcoded passwords that add

to security cracks. Passwords are relatively weaker; even if regularly updated, infiltration is more manageable [34,35].

A resource constraint is also attached to IoT devices that limits their computing capacity to execute more robust security protocols. Advanced security settings are missing in many IoT devices. For example, humidity and temperature sensors cannot undertake advanced encrypted settings or any more robust security measures. Moreover, IoT devices are devoid of security upgrades or patches throughout their life cycle. As far as the manufacturers' viewpoint is concerned, installing advanced security can increase product cost, stall its development, and hamper its proper functioning [36].

The server–client model serves as the basis for most IoT devices being used today, whereby identified and authenticated devices are connected across cloud servers, possessing considerable processing power and an enormous range of storage capacity. Each connected device is connected through the internet, regardless of the device's proximity. It requires a significant number of communication linkages to be formed, extensive networking of devices, and maintenance of centralized clouds [34]. These features stretch the cost for large IoT networks to a great extent. Moreover, the dependency of the whole setup on cloud servers makes the entire model vulnerable to a single-point failure. It should be assured that IoT nodes are secure from any sort of physical meddling or data breaches. Many techniques to safeguard IoT devices exist, but most are too complex and unsuitable for IoT devices with resource constraints and restricted computation capacity [37,38].

### 2.4. Blockchain (BC)

BC is an immutable ledger that files data records in a decentralized way. It replaces the need for a mutually trusted central third party and allows entities to engage securely. Blocks of data are maintained by the BC, which keeps hold of ever-growing sets of data entries. If accepted by the BC, these data blocks are connected to past and future data blocks via cryptographic protocols. These data blocks can be written in the BC, read, and tamper-proofed by participating entities using a consensus mechanism [39]. This feature enables decentralization in data management and related transactions [40,41]. Figure 3 provides a sketch of the basic operation of BC.

Moreover, BC eradicates dependency on central authority and facilitates self-executing smart contracts. Ethereum is the most significant proponent of the smart contract using BC [32]. Table 2 provides a brief outline of the required features of BC in the case of EHR applications [42] 5s [34,35,43]. Characteristics of BC have been summarized in Figure 4.

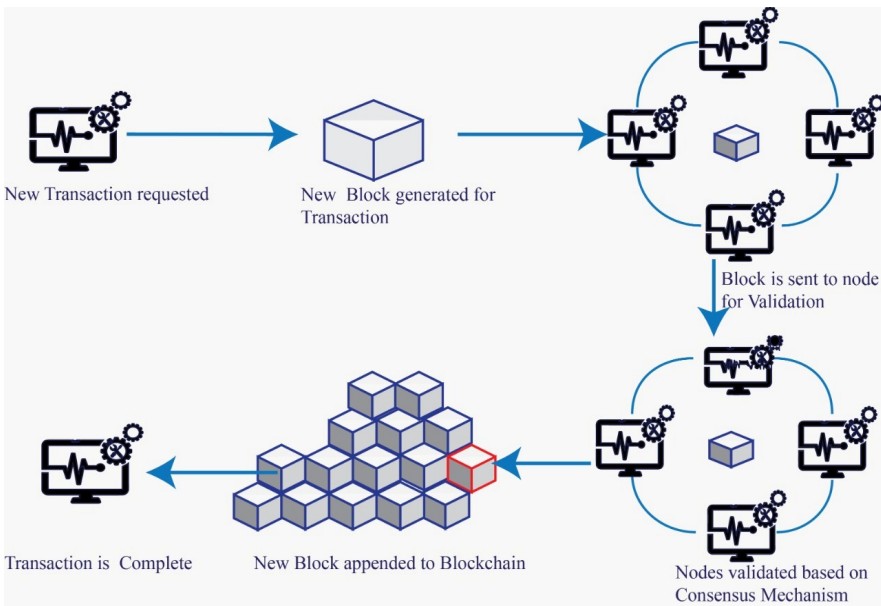

**Figure 3.** Basic operation of blockchain.

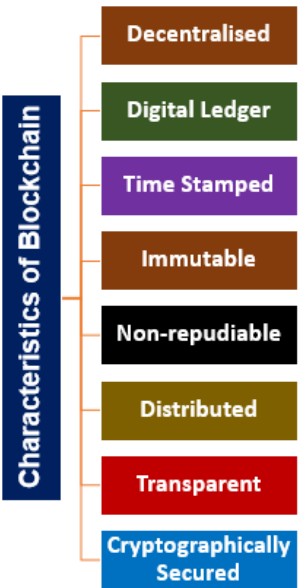

**Figure 4.** Characteristics of blockchain.

**Table 2.** Required features of BC in the case of EHR.

| Feature | Description |
| --- | --- |
| Decentralization | BC allows for decentralization, which makes storing crucial data like documents, contracts, etc., easier, which means the possessor can access it from everywhere through the net. Moreover, full account control is in their hands, and they can share their data/assets with any entity they need to. |
| Transparency | The details of assets and transactions are made public and can be viewed by all parties, allowing for maximum transparency. |
| Immutability | No entity member can change the data once fed into the BC ledger. In case of an error, a new transaction must be made for rectification. However, both transactions, the erroneous and the rectified, will be shown in the ledger. |

Types of BC

BC can be mainly divided into three groups based on the consensus mechanisms. These three kinds of BC are explained more below.

- Public BC: Anybody can join and exit the BC network. A participant does not need authorization to function as a miner or a typical BC node, and everyone has equal access. Incentives are used to guarantee participants' involvement and activity in such a BC.
- Consortium BC: It allows just a select set of nodes to act as the governing authority in the consensus mechanism.
- Private BC: A specific entity manages, approves, and administers a private BC. Users must obtain permission from the proper authorities in order to participate. Transactions are confirmed in confidence and may not be available to the general public. A private BC frequently generates blocks faster and produces more transactions than other types of BC.

### 2.5. Blockchain and IoT in EHR

The IoT enables connectivity to everyday gadgets and other devices using the internet. The internet connection, electronics, and other hardware inputs allow these devices to interact over the internet. Remote monitoring and control can also be conducted on these devices [44]. BC aptly complements the rigid settings required by IoT networks in the following ways:

- It offers a secure platform whereby communication can safely take place between all devices connected to the network.
- It provides for ample security of the network, which safeguards the stored data against any information attacks.

2.5.1. Benefits of BC and IoT application in EHR

The application of BC and IoT in EHR systems has many apparent advantages that have been summarized with the following points:

a. Privacy/Anonymity: BCs employ public-key cryptography and use digital identities specific to various transactions. This feature obscures the actual identification of IoT applications that withhold sensitive information [44].

b. Smart Contracts: Smart contracts are those that are executed once their conditions are fulfilled. Certain BCs like Ethereum provide this facility. For instance, one end of the system can make payments when certain associated conditions are fulfilled, like some product/service being delivered [45].

c. Auditability: Auditability is a crucial part of security. It actively records in the audit logs who is accessing what information, through which system, and for what purpose. It also ensures the time stamping of each operation conducted during all phases of its lifecycle [46,47].

d. Trustworthiness: The feature of data sharing of IoT applications across an infrastructure that is under the control of numerous organizations upholds trustworthiness. This sharing is essential for enhancing the performance of services offered by these organizations [48,49].

e. Security:

  ■ Privacy: Only allows authenticated members to gain access to stored data. To preserve confidentiality and complete privacy, blockchain applications must be used wisely with other cryptographic mechanisms [50].

  ■ Integrity: An unidentified entity cannot modify the recorded data. It is a must that the data being transmitted are accurate.

  ■ Availability: The access to information is levied to legitimate users, and any improper access denial(s) to resources is prevented.

  ■ Accountability: Every requisite individual or entity will be duly audited, supervised, and held accountable for any adversity.

f. DDoS warning and Mitigation: BC and smart contracts can merge together in collaborative architectures that can produce DDoS notifications on numerous domains. With transactions based on BC, it becomes improbable for information attackers to launch malware on devices connected through the IoT network and install their IoT botnets to make DDoS attacks. The stringent check on outgoing traffic makes it impossible for DDoS messages to spread from IoT devices [42].

2.5.2. Blockchain application in IoT-based EHR

BC is used in IoT–EHR to integrate the IoT sensors to provide secure communication and storage of health records. Various researchers have proposed BC for IoT-based EHR. These research outcomes have been summarized in this section and presented in Table 3. The framework for BC-based storage of data generated through IoT-based healthcare equipment was proposed by [51] and guaranteed patient safety. The proposed framework contains a virtual patient agent (PA), specifying the capabilities of BC.

**Table 3.** Blockchain applications in IoT-EHR.

| Ref. | Framework | BC Type | Consensus Mechanism | Type of Storage | Smart Contract | Domain |
|---|---|---|---|---|---|---|
| [51] | Not Defined | Consortium | Verified by group head, and then blocks are added | Off-chain (On cloud) | Data management and analysis | IoMT data management |
| [52] | Not Defined | Private | Not Defined | On-chain (hospitals) | Not Defined | Data Security |
| [53] | Ethereum | Private | Not Defined | Off-chain (exterior server) | Smart health records illustration | IoMT data management |
| [54] | Ethereum | public | Not Defined | Off-chain (IPFS) | Manage patients and doctor's communication | IoMT data management |
| [55] | Ethereum | Private | Proof of medical stake | Off-chain (IPFS) | Manage access control | Access control in IoMT |
| [56] | Hyperledger Fabric | Private | Not Defined | On-chain | Verification and validation of transactions | Remote health monitoring |
| [57] | Not Defined | Consortium | BFT-SMaRt | On-chain | Not Defined | IoMT data management |
| [58] | Not Defined | Private | Verified by Cluster head, and then blocks are added | Off-chain (cloud) | IoMT data analysis and patient health monitoring | Patient monitoring remotely |
| [59] | Ethereum | Private | Not Defined | Off-chain (on cloud) | Access control | Monitor a neurological disorder of patients |

In [52], the authors have used various security and identity disclosure terms from BC technologies when exchanging patient information through IoT devices. The hashing technology is used to encrypt transactions with confidential and critical patient data using a new encryption algorithm. The BC advantages of secure and reliable storage and IoT medical system data sharing with patients and healthcare providers have been discussed in [54]. The patient information is recorded in the BC, while the IoT medical system data are recorded in external databases such as IPFS. Smart contracts are being utilized to assure privacy and confidentiality.

An IoT BC-based architecture was suggested in [56] to enable remote patient monitoring. Transactions can be checked and verified with smart contracts to be carried out by peers supporting the Byzantine Fault Tolerance algorithm. In [58], a custom IoT medical device BC platform has been suggested. First, the proposed BC is private; nodes should be allowed for network connectivity and transmission. Second, the authors delete the power of work (PoW) consensus protocol. MedChain [57] is a consortium-based BC platform suggested to solve the complexities of safely recording the data blocks generated by medical devices. It involves processing time-series data sources, maintaining immutable and unalterable medical records, and facilitating effective storing and exchanging of massive and critical information.

A private Ethereum-based infrastructure to execute smart contracts for user/device requests and track access, consisting of a set of credentials, location, and domain attributes, is proposed by [55]. The IPFS was employed to store information on the patient's personal health and IoT devices. A private BC-based healthcare data management system

was proposed by [53]. It runs on Ethereum-based smart contracts to control data access authorization for organizations such as patients, clinics, physicians, research institutions, and other participants. In [59], the authors designed a cloud-based framework to track neurological disorder progression using IoT medical devices. It uses cloud storage to record and process IoT medical device information and incorporates Ethereum BC to share and transfer information securely between health facilities and users. A general framework of BC and IoT-based EHR systems has been presented in Figure 5 for reference.

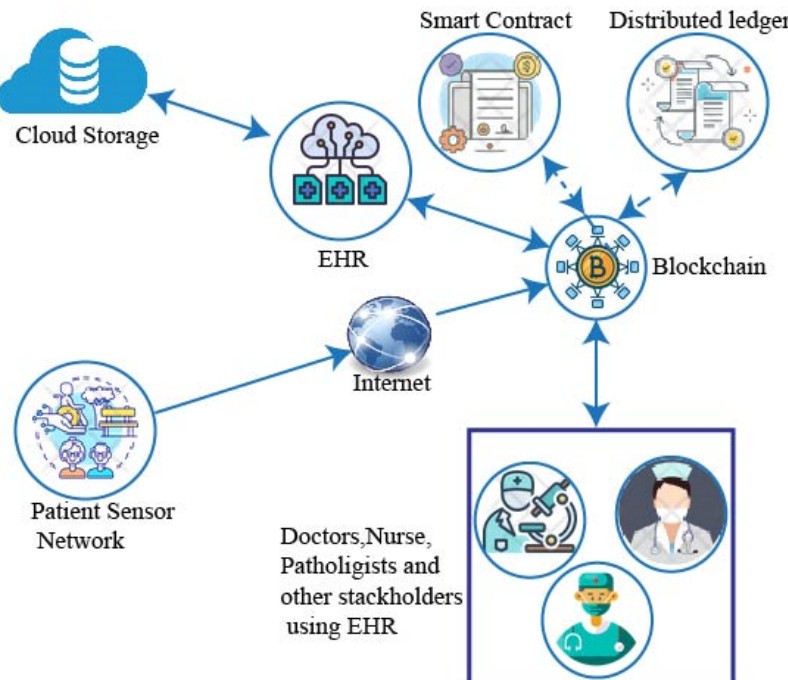

**Figure 5.** General framework of blockchain and IoT-based EHR system.

## 3. Comparative Study of Blockchain Consensus Mechanism for IoT-Based EHR

A consensus mechanism is a primary criterion to appraise the efficiency of any BC-based system. Many consensus mechanisms are available for a BC-based system, but these cannot be used in IoT-based EHRs, as the resource requirements differ. Many key consensus algorithms are described below which can be used in various ways, particularly e-healthcare services [60]. Table 4 provides a study of BC consensus algorithms for IoT-EHR.

i.      PoW (Proof of Work): It is based on the computational effort required based on mathematical puzzles used in asymmetric cryptography. Solving a problem is complex, but verifying that output is easy. As PoW is widely used in several platforms, due to high complexity and resource requirements, there is a mild prospect of involving PoW in healthcare systems involving IoT devices [61].

ii.     LPoS (Leased Proof of Stake): Addresses centralization in PoS, makes low-balance nodes, leases contracts, and shares benefits with the owner. The PoS consensus algorithm will facilitate a high-quality e-health service [62].

iii.    DPoS (Delegated Proof of Stake): With DPoS developed from PoS, network users can elect delegates to verify blocks. It can be used in highly possible electronic health situations [63].

iv.     PoI (Proof of Importance): It is an enhancement of PoS. It studies the nodes' balance and nodes' credibility. It is an efficient network. We suggest using it for e-healthcare systems, as healthcare professionals' credibility may be used for patient decision making [64].

v.    PBFT (Practical Byzantine Fault Tolerance): Each node works together to add the next block. Consensus requires 2/3 nodes. It provides low tolerance to malicious nodes. It is recommended for healthcare use [65].

vi.    PoA (Proof of Activity): It is a hybrid version of PoW and PoS. First, PoW is completed. Then, after a PoS, a group of verifiers sign jointly to place the transaction in the miner's header. Despite the long delay, it is not ideal for IoT; therefore, e-healthcare is not a reasonable option [66].

vii.    DBFT (Delegated Byzantine Fault Tolerance): It is an enhancement of PBFT. Nodes are selected as representatives of another node. Therefore, using dBFT in IoT-based BC healthcare frameworks is not fully understood [67].

viii.    PoC (Proof of Capacity): It is an upgraded PoW. It is used to record large data sets for mining other nodes' next blocks. It is not adequate for IoT but is used for other health-specific programs [68].

ix.    PoS (Proof of Stake): A prevalent consensus mechanism randomly selects the node to tackle and which block to mine next. Within PoS, the mining reward/coin production does not exist, but the miner is compensated with a transaction fee [69].

x.    PoB (Proof of Burn): It sends coins to an irreversible address. Many burned coins support miners in mining. It is a good choice for cryptocurrency architecture but poor for IoT due to the entirely conditional economic model and burning of the coin. Because of its uncontrolled burning method, it is not appropriate for e-healthcare applications [70].

xi.    Proof of Trust (PoT): A consensus algorithm that offers equal opportunities to participate in crowdsourcing activities. Owing to the difference in reputational standards, only a few members are not in the consensus nodes. PoT consensus uses subjective logic algorithms, using time signs and digital signatures to maximize block node unpredictability. The improved algorithm will automatically complete a reputation evaluation of participating crowdsourcing members. The POT can achieve validity, fairness, and security [61,71].

xii.    Proof-of-Luck (PoL) consensus algorithms execute the real-time protocol for the Gateway Agreement [63]. It provides IoT data tolerance and generates encryption digests for input validation. It uses SHA-256 to build replicated data digests [72].

**Table 4.** Comparative analysis of blockchain consensus mechanism for IoT-HER.

| Algorithms | CHARACTERISTICS | | | | | | |
|---|---|---|---|---|---|---|---|
| | IoT Compliant | Basic Concept | Popularity | E-Health Support | Adaptability | Accessibility | Energy |
| PoW [61] | ○ | CPU | ● | ◑ | ● | Open | ● |
| LPoS [62] | ◑ | PoS | ◑ | ● | ◑ | Open | ◑ |
| DPoS [63] | ◑ | PoS | ◑ | ● | ◑ | Open | ◑ |
| PoI [64] | ○ | PoS | ◑ | ● | ◑ | Open | ◑ |
| PBFT [65] | ○ | 67% Node | ○ | ● | ○ | Prop | ○ |
| PoA [66] | ○ | PoW-PoS | ○ | ○ | ◑ | Prop | ○ |
| DBFT [67] | ○ | PBFT | ○ | ○ | ○ | Prop | ○ |
| PoC [68] | ○ | PoW | ○ | ○ | ○ | Open | ○ |
| PoS [69] | ◑ | Stake | ● | ● | ● | Open | ◑ |
| PoB [70] | ○ | – | ○ | ○ | ○ | Prop | ○ |
| PoT [71] | ● | PoW | ◑ | ● | ◑ | Prop | ○ |
| PoL [72] | ◑ | PoW | ◑ | ● | ◑ | Prop | ○ |

*Note:* ◑—medium/partial, ●—High, ○ Low/No.

## 4. Blockchain-Based Framework for IoT-EHR

We have proposed an IoT-based EHR system with BC integration which can provide these benefits over the traditional healthcare system:

1.  Privacy and tracking of EHR of IoT-based patients without alteration or corruption;
2.  Security of EHR data is assured;
3.  To give and revoke permission by patients to parties wishing to use the EHR data;
4.  It provides a framework for engaging numerous healthcare organizations and pharmaceutical companies in clinical trials and research on drug design, medications, and delivery facilities across the publicly accessible ledger database;
5.  It reduces operating costs and increases interoperability, universal accessibility, and truthfulness.

The proposed BC-based EHR framework supports the integration of IoT devices into EHR. It can be further upgraded to integrate with other healthcare facilities requiring unified integration of personal health records and monitoring of patients. This framework has been presented in Figure 6, consisting of three main layers of participants. These are:

A.  EHR layer: At the EHR layer, which can also be termed as the healthcare provider layer, different healthcare organizations and entities collaborate to share their specific healthcare records, irrespective of the EHR storage type.
B.  BC layer: This layer connects with the EHR layer with the help of an interface that translates the records into a unified format, and details are stored in IPFS storage to support interoperability. The BC layer comprises a smart contract, storage policy, EHR manager, consensus mechanism, and IPFS storage. The EHR manager manages the records from different EHRs and processes them. The PoT consensus mechanism processes the new records before storing them in BC. Smart contracts provide auto-execution required for transaction processing.
C.  IoT-based patient monitoring layer: The patient sensor layer consists of different sensors to sense the various inputs for the patients, such as BP, EMG, ECG, glucose level, etc.
D.  User layer: The users connect with the system using the interface. They can enter any new record or view them based on their authorization in a standard template, irrespective of their actual storage format.

The following general steps are followed during the process:

1.  Different hospitals or service providers can have their EHR with a heterogeneous structure having health records. They are processed at the BC layer for the sake of interoperability and security.
2.  The EHR layer is connected with the BC layer, and all the authentication and verification of records are conducted at the BC layer before storing them in EHRs. There is no direct connection between users of the system and EHRs.
3.  The IoT layer collects patient data. These sensor data are passed to the BC layer via the IoT server and are further processed using smart contract and storage policy before storing in the data storage.
4.  The BC layer consensus mechanism will mine and store the newly sensed data in the IPFS storage.
5.  The old records are mined using the EHR layer and processed. Furthermore, their hashes are stored in the BC layer to marinate records' immutability.

The identical copy of the EHR is stored in IPFS storage at the BC layer and EHR storage of individual organizations. It also helps in achieving the interoperability of records.

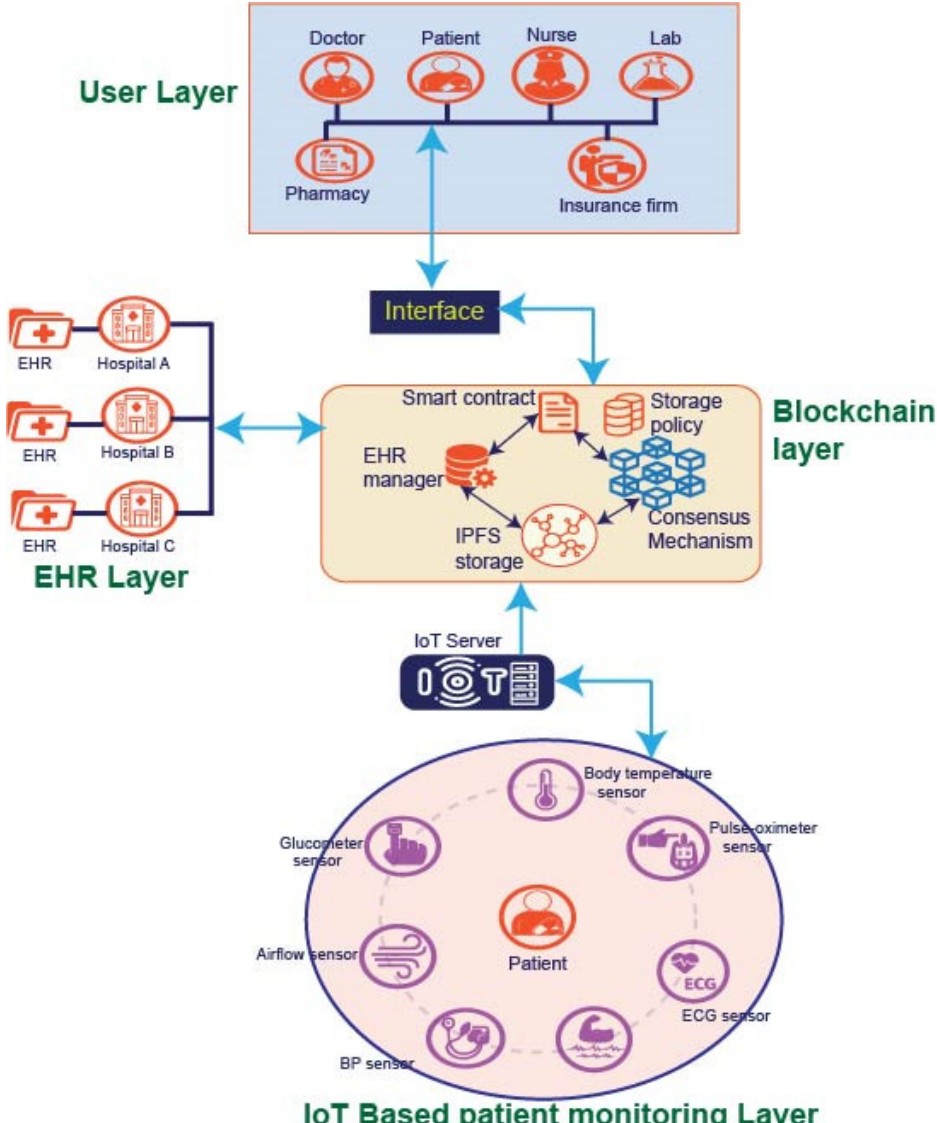

**Figure 6.** Blockchain-based IoT-EHR framework.

Algorithm 1 defines the sample algorithm to show the pattern of monitoring the vital signs of patients. In this algorithm, we have discussed the monitoring of oxygen saturation level in the patient's body, which will raise the alarm when the oxygen level goes below 94%.

| **Algorithm 1:** Oxygen Saturation Analysis |
|---|
| 1:   **oxygen_R   Read Oxygen Saturation from sensor** |
| 2:   **Procedure oxygen_sat ()** |
| 3:   **Store   False** |
| 4:   **if (oxygen_R $\geq$ 94) then** |
| 5:   **Store True** |
| 6:   **end if** |
| 7:   **oxygen_sat   Store** |
| 8:   **end procedure** |

Algorithm 2 shows the pattern of adding new EHRs into the database that will be added to the blockchain using a proper consensus mechanism. We have discussed and evaluated the different consensus mechanisms and summarized them in Table 4. It is

evident from the analysis that the PoT consensus mechanism is most suitable for IoT requirements; hence, it will be used for finally adding the EHRs to the blockchain.

| **Algorithm 2:** Load record in EHR |
|---|
| 1:   **Load_EHR Store Record in EHR** |
| 2:   **Procedure EHR ()** |
| 3:   **If Key_Entry == Owner_Key then** |
| 4:   **Create health Record object** |
| 5:   **Push the object in EHR** |
| 6:   **Return "New Record Stored"** |
| 7:   **Else** |
| 8:   **return Not authorized** |
| 9:   **end if** |
| 10:  **end procedure** |

Algorithm 3 shows the mechanism for retrieving and viewing the EHR data. If the applicant is a medical practitioner/doctor, they can access all the attributes (details) of the patients. Furthermore, other classes of users can access limited records or characteristics from the EHR. Algorithm 3 can be further extended based on the different types of users. The algorithms mentioned above only show a sample of the system working.

| **Algorithm 3:** Read EHR data |
|---|
| 1:   **View EHR record** |
| 2:   **procedure View EHR ()** |
| 3:   *If Applicant* $\in$ **Doctor then** |
| 4:   **Include All EHR Features** |
| 5:   **return EHR string** |
| 6:   **else if** *applicant* $\in$ **Other then** |
| 7:   **Only Include EHR Attributes accessible by a Users** |
| 8:   **return EHR string** |
| 9:   **else** |
| 10:  **return Not authorized** |
| 11:  **end if** |
| 12:   **end procedure** |

*Security Analysis of the Proposed Framework*

The proposed framework is based on the BC platform, providing many inherent security features. In this section, a theoretical security analysis of the proposed model has been presented. The suggested model has been assessed in terms of privacy, data integrity, availability, and access control.

a.  Privacy: BC's main strength lies in its immutability feature. The records are stored in a decentralized manner, and elliptical curve cryptography (ECC) is used to secure against privacy breach attacks and single-point failure. The decentralized nature also makes it secure against man-in-middle attacks.

b.  By utilizing digital signatures and the blockchain approach, the proposed solution significantly upholds the confidentiality of the data. The next step is to request permission to access the health archive's record on the patient's health. As a result, a session key is provided to the doctor so that they can access the EHR. This key allows access to the data and establishes the patient's identity. A variety of degrees of authentication are employed to protect data confidentiality.

c.  Data Integrity: The hash of each record is stored, which can be used to verify the integrity of individual transactions. Each transaction is appended to the BC utilizing a consensus algorithm, but most existing consensus mechanisms are unsuitable for

resource-constrained IoT devices. PoT can be a suitable consensus mechanism choice in such a case.

d.  In order to learn more about their clinical knowledge, doctors and patients want access to their EHRs. The client must first receive approval from the EHR system's repository. The user information on the access list is double-checked to make sure of this. The customer is given access to the record at the stage at which they have been allowed access, if the value matches.

e.  Availability/DDoS Attack: Availability attacks affect resource or system usability. DDoS is a primary availability attack against any IoT network originating from unauthorized requests from unknown nodes. Such attacks can be avoided if nodes are fully identified and authorized. Two-factor authentication using IoT server and client IoT devices can be used to counter such attacks. BC-based identity frameworks can be other possible solutions.

f.  Authentication and Access Control: Proper authentication and role segregation of entities are essential for access control. The smart contract-based approach in the proposed framework provides a solution for proper access control.

The re-encryption key is created using the user's private key and the keyword. Only another doctor's public key can decrypt the EHR ciphertext that has been saved in a specific location and encrypted. Additionally, a patient's private key and keyword are the only ways for an authorized third party to decipher a particular ciphertext.

## 5. Future Work

The general challenges of applying BC in IoT-based EHRs require further investigation and research.

a.  Resources constraint: IoT systems have restricted memory and processing capacity, while BC requires tremendous energy. BC's computational specifications for mining blocks are far beyond resource-constrained IoT devices.

b.  Bandwidth constraint: Verification of transactions is facilitated by the decentralization of the BC, where network nodes work together. The bandwidth of IoT devices in the end-device layer is constrained. BC-based applications may require more bandwidth; thus, any edge device should be able to handle them.

c.  Connectivity constraint: All nodes remain attached to the BC and communicate through predetermined protocols within BC technology. This feature also connects BC to IoT devices and is perhaps more vulnerable to security threats.

d.  Memory constraint: Many public BC technologies start charging transaction fees and use them to compensate those peers engaged in block mining. However, in the case of healthcare software, our requirements and limitations are very exceptional. Health data are analyzed regularly. Collecting and storing health data for various patients could expose a severe memory issue.

e.  GDPR compliance: GDPR mandates the appropriate and transparent acquisition, processing, and storage of personal data to reclaim data control. GDPR makes data protection compliance more manageable and less expensive for businesses. GDPR and HIPAA are primarily used to reduce the likelihood of privacy abuses in healthcare data [73].

## 6. Conclusions

This study analyzes various IoT, EHR, and BC technological fundamentals. Recent studies fusing all three technologies in the healthcare field have been reviewed. It studies the security concerns and difficulties in the IoT-EHR. The discussion of BC technology and its potential application in easing security concerns associated with IoT-EHR integration has been developed. It examines the various technological, security, and consensus techniques for BC to address security issues. The article reviews the potential applications of BC and IoT to improve electronic health records and other e-healthcare services. A suitable consensus mechanism for IoT-EHR systems is vital for providing efficient and

secure BC-based IoT-EHR systems. It is the core of BC operation, and it is also the most resource- and energy-consuming part. IoT-based systems are typically unable to handle such a high resource requirement. Major BC consensus methods that might be employed in IoT-based EHR systems have been analyzed on defined parameters. Based on the review, the PoT consensus mechanism may be the most suitable for IoT-EHR systems out of the mechanisms evaluated, as it can support health applications, provide sufficient security, and consume less energy. The paper further suggested a new BC-based IoT-EHR framework for processing and retrieving EHR records securely and reliably while maintaining interoperability characteristics. A theoretical security analysis of the framework has been provided to support the suggested framework's security on several security parameters like privacy, integrity, availability, authentication, and access control. A more workable consensus mechanism that meets the needs of IoT-EHR and how the IoT-EHR systems can completely comply with GDPR, along with other future research directions, have been highlighted in this research work.

**Author Contributions:** Conceptualization, S.A. (Shadab Alam), M.M.K., and M.S.; formal analysis, S.A. (Shadab Alam) and S.B.; investigation, M.S., S.A. (Sadaf Ahmad), S.B., and M.M.K.; project administration, S.B. and A.A.M.; resources, F.A. and A.A.M.; supervision, S.B. and A.A.M.; visualization, F.A. and M.M.K.; writing—original draft, S.A. (Shadab Alam) and S.B.; writing—review and editing, M.S., F.A., M.M.K., and A.A.M. All authors have read and agreed to the published version of the manuscript.

**Funding:** This research was funded by Princess Nourah bint Abdulrahman University Researchers Supporting Project number (PNURSP2023R151), Princess Nourah bint Abdulrahman University, Riyadh, Saudi Arabia.

**Institutional Review Board Statement:** Not applicable.

**Informed Consent Statement:** Not applicable.

**Data Availability Statement:** Not applicable.

**Conflicts of Interest:** The authors declare no conflict of interest.

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
