# Peer review of "An Overview of Blockchain and IoT Integration for Secure and Reliable Health Records Monitoring"

_sustainability, doi:10.3390/su15075660_

Round 1
Reviewer 1 Report
1: There is no discussion wrt to concrete literature, as to why BC solutions are good wrt to existing cryptographic mechanism like ABE etc. for example (Granular data access control with a patient-centric policy update for healthcare). Existing relevant papers on IoT based healthcare with existing cryptographic techniques should first be mentioned and explained as to why BC based approach is more appropriate and best. The issues in existing schemes should be highlighted if any. 2: Although several BC consensus mechanisms and types of BC is discussed. However, no discussion exists of the pro and cons of these, and which one will be adopted by authors in Section 4. 3: Out of 65 cited papers, only 5 are from 2022, out of which 3 are survey papers. More technical papers based on BC based IoT healthcare should be added and discussed. for example (FBASHI: Fuzzy and Blockchain-Based Adaptive Security for Healthcare IoTs) 4: Regarding Section 4, there seems no contributions. Fig 6 cannot be related directly to text in this sections. This should be revised with context of point 2. 5: No simulation results are presented, as to how it can be evaluated? Even if performance evaluation is not presented, some how it should be compared based on relevant metrics with existing works etc. 6: Intrinsically BC does not provide confidentiality, however, with access control limited to certain entities, it can be enforced. There are some contradictions regarding it through out the paper.Author Response
Thankyou for your comments.
|
Reviewer 1 |
|
|
Reviewer Comment |
Response to Reviewer's Comments |
|
1: There is no discussion wrt to concrete literature as to why BC solutions are good wrt to existing cryptographic mechanism like ABE etc. for example (Granular data access control with a patient-centric policy update for healthcare). Existing relevant papers on IoT based healthcare with existing cryptographic techniques should first be mentioned and explained as to why BC based approach is more appropriate and best. The issues in existing schemes should be highlighted if any. |
Some more relevant literature on IoT applications in healthcare has been added in the Introduction section that is not applying the Blockchain. Further, the issues of general IoT-EHR systems have been highlighted, and how Blockchain can help resolve these issues has been presented in the Introduction section. Table 1 has also been thoroughly updated accordingly. |
|
2: Although several BC consensus mechanisms and types of BC is discussed. However, no discussion exists of the pro and cons of these, and which one will be adopted by authors in Section 4. |
The given BC consensus mechanisms have been evaluated on the parameters of IoT Compliant, Basic Concept, Popularity, E-Health support, Adaptability, Accessibility and Energy consumption that has been presented in Table 4. Our main aim in discussing these consensus mechanisms is to evaluate which consensus mechanism will be best suitable for IoT-based applications and in the healthcare domain. Section 4 has been fully revised as per the suggestions of the review comments and alignment with the consensus mechanism and Figure 6. Figure 6 has also been updated for better clarity. |
|
3: Out of 65 cited papers, only 5 are from 2022, out of which 3 are survey papers. More technical papers based on BC based IoT healthcare should be added and discussed. for example (FBASHI: Fuzzy and Blockchain-Based Adaptive Security for Healthcare IoTs) |
We have added some of the latest references in the introduction section and in Table 1 in line with the review comments. The article "FBASHI: Fuzzy and Blockchain-Based Adaptive Security for Healthcare IoTs" has also been included in the study |
|
4: Regarding Section 4, there seems no contributions. Fig 6 cannot be related directly to text in this sections. This should be revised with context of point 2. |
Section 4 has been fully revised as per the suggestions of the review comments and alignment with the consensus mechanism and Figure 6. Figure 6 has also been updated for better clarity. |
|
5: No simulation results are presented, as to how it can be evaluated? Even if performance evaluation is not presented, some how it should be compared based on relevant metrics with existing works etc. |
The proposed framework has been evaluated for its security in section 4.1. Implementation is beyond the scope of this work. |
|
6: Intrinsically BC does not provide confidentiality, however, with access control limited to certain entities, it can be enforced. There are some contradictions regarding it through out the paper. |
The issue has been identified and corrected. In some places, it has been written with a sense of privacy. Now it has been written as privacy instead of confidentiality. BC deals with the privacy aspects in collaboration with other cryptographic techniques that have been further highlighted and cited. |
Reviewer 2 Report
The contribution of the paper is very trivial.
The content of the paper is a survey on BC, Iot, and EHR.
No BC practical implementation, No model, No analysis for measuring for any security services.
Author Response
Dear Reviewer
Thankyou for your comments.
|
Reviewer 2 |
|
|
Reviewer Comment |
Response to Reviewer's Comments |
|
The contribution of the paper is very trivial. The content of the paper is a survey on BC, IoT, and EHR. No BC practical implementation, No model, No analysis for measuring for any security services. |
The model for Blockchain-IoT integration has been presented in section 4 and given in figure 6. The security analysis has been given in section 4.1. Implementation is beyond the scope of this work. |
Reviewer 3 Report
The paper contributes IoT, and BC scheme for EHR, which enables ease security concerns associated with IoT-EHR integration. The topic and research are interesting and meaningful. However, there are some problems that should be addressed before it is considered for publication.
1), the figures in your paper are a bit blurry. Please consider replacing them with clearer ones.
2), there is some statement confusions in the manuscript, such as, in the summary on page 14, I don't understand what it means by the first sentence. Please check the manuscript carefully. The manuscript needs careful editing and particular attention to English grammar, spelling, and sentence structure.
3), The writing is not clear enough. It's too vague. For example, the framework described on page 12 does not match figure 6. The authors need to refine figure 6 so that reviewers can understand the framework more clearly.
4), The significance of this paper is not expounded sufficiently. The authors need to highlight this paper's innovative contributions.
Author Response
Dear Reviewer
Thankyou for your comments.
|
Reviewer 3 |
|
|
Reviewer Comment |
Response to Reviewer's Comments |
|
1), the figures in your paper are a bit blurry. Please consider replacing them with clearer ones. |
The images have been updated with high resolution. |
|
2), there is some statement confusions in the manuscript, such as, in the summary on page 14, I don't understand what it means by the first sentence. Please check the manuscript carefully. The manuscript needs careful editing and particular attention to English grammar, spelling, and sentence structure. |
There were some typos and grammar issues. The manuscript has been thoroughly revised to correct English grammar, spelling, and sentence structure errors. |
|
3), The writing is not clear enough. It's too vague. For example, the framework described on page 12 does not match figure 6. The authors need to refine figure 6 so that reviewers can understand the framework more clearly. |
Section 4 has been fully revised as per the suggestions of the review comments and Figure 6. Figure 6 has also been updated for better clarity. |
|
4), The significance of this paper is not expounded sufficiently. The authors need to highlight this paper's innovative contributions. |
This article's contribution has been defined better under the Introduction and conclusion sections. The Introduction and Conclusion section has been thoroughly revised to highlight the innovative contributions of the article. |
Round 2
Reviewer 1 Report
I do not feel significant contributions being made to this paper in the revision.
Section 4 just descriptively details IoT EHR but no concrete details.
No performance comparison / security analysis etc. indicative of the effective to the proposed approach.
The authors should revisit previous comments and make revisions effectively.
Author Response
|
Reviewer 1 |
|
|
Reviewer Comment |
Response to Reviewer's Comments |
|
I do not feel significant contributions being made to this paper in the revision. |
All the suugested changes have been incorporated for the previous suggestions. Also section 4 has been further updated. |
|
Section 4 just descriptively details IoT EHR but no concrete details. |
In the section 4, we have further added the 3 sample algorithms for IoT sensor monitoring, EHR record addition and procedure to view EHR record for authorized users. |
|
No performance comparison / security analysis etc. indicative of the effective to the proposed approach. |
The security analysis in the section 4.1 has been elaborated and detailed security analysis has been provided. |
|
he authors should revisit previous comments and make revisions effectively. |
We have incorporated all the previous comments and also incorporated the new suggestions given in this revised manuscript. |
Reviewer 2 Report
The paper presents a review study on BC, and EHR system.
This review study have been done on many research, so this study did not present any significant contribution in comparison with the others.
This paper is not reasonable for publication in its current form.
The paper could be enhanced by adding a case study with implementing practical parameters and measuring sensible outcomes.
Author Response
|
Reviewer 2 |
|
|
Reviewer Comment |
Response to Reviewer's Comments |
|
The paper presents a review study on BC, and EHR system. |
Yes it is true but furher a framework has been proposed that has been analysed, the security analysis has been presented as well as comparative analysis of different consensus mecyhnaisms have been given to identify most suitable blockchain consensus mechanism for IoT based systems. |
|
This review study have been done on many research, so this study did not present any significant contribution in comparison with the others. |
The comparison of our work has been given in the Table 1. |
|
This paper is not reasonable for publication in its current form. |
We have revised the manuscript further mainly in section 4. |
|
The paper could be enhanced by adding a case study with implementing practical parameters and measuring sensible outcomes. |
In the section 4, we have further added the 3 sample algorithms for IoT sensor monitoring, EHR record addition and procedure to view EHR record for authorized users. The security analysis in the section 4.1 has been elaborated and detailed security analysis has been provided.
|
Reviewer 3 Report
1) The research design, questions, hypotheses and methods should be stated more clearly.
2) Many figures in the revised version are still not clear enough, which should be fixed before publication.
3) The language and style of the current version of the manuscript can be improved.
Author Response
|
Reviewer 3 |
|
|
Reviewer Comment |
Response to Reviewer's Comments |
|
1) The research design, questions, hypotheses and methods should be stated more clearly. |
These aspects have been improved |
|
2) Many figures in the revised version are still not clear enough, which should be fixed before publication. |
Figures 1, 3 and 4 have bene again updated for better clarity. |
|
3) The language and style of the current version of the manuscript can be improved. |
The manuscript have been further improved and corrected for improving the language issues. |
|
|
|
Round 3
Reviewer 1 Report
The authors have addressed the concerns of the reviewer
Author Response
Dear Reviewer
We sincerely appreciate your effort in reviewing this manuscript and giving contributions that will increase the quality of this manuscript. Also, we appreciate the Editor for enabling the platform to send out papers for possible publication.

Reviewer 2 Report
I prefer adding a case study to show how these thoughts can be implemented in a practical form, otherwise the title of the paper must be changed to " An overview on blockchain and IoT integration......"
Author Response
Dear Reviewer
We sincerely appreciate your effort in reviewing this manuscript and giving contributions that will increase the quality of this manuscript. Also, we appreciate the Editor for enabling the platform to send out papers for possible publication. We have changed the title as per your comments to “An overview of Blockchain and IoT integration for Secure and Reliable Health Records Monitoring”
